# Antibiotic Resistance Genes Occurrence in Conventional and Antibiotic-Free Poultry Farming, Italy

**DOI:** 10.3390/ani12182310

**Published:** 2022-09-06

**Authors:** Muhammad Farooq, Camilla Smoglica, Fausto Ruffini, Lidia Soldati, Fulvio Marsilio, Cristina E. Di Francesco

**Affiliations:** 1Faculty of Veterinary Medicine, University of Teramo, Loc. Piano D’Accio, 64100 Teramo, Italy; 2Gesco Cons. Coop a r.l., 64020 Teramo, Italy

**Keywords:** broiler, antibiotic-free, antibiotic resistance genes (ARGs), PCR, *tet*(M)

## Abstract

**Simple Summary:**

Antibiotic resistance represents an emergent threat for animals, humans, and the environment. Food-producing animals are directly involved in the dissemination and maintenance of resistant bacteria and their genetic determinants. The genes responsible for resistance mechanisms in bacteria can be located at the chromosomal level or in extra-chromosomal mobile genes, and are transferrable to other bacteria by different modalities. The use of antibiotics in livestock can exercise a selective pressure on resistance determinants and their persistence in intestinal microbiota. In this respect, poultry farming represents a potential driver of resistance genes as these genes can reach the environment or the food chain from poultry litter. In this study, litter samples from conventional flocks (where antibiotics are allowed only for therapeutic purposes) and antibiotic-free farms (where these molecules are excluded) are screened for some resistance genes against antibiotics routinely used in veterinary practice (tetracyclines, aminoglycosides, lincomycin), along with some critically/highly important antibiotics exclusively intended for humans (chloramphenicol, colistin, vancomycin, carbapenems). The results showed the presence of several resistance genes in poultry litter from both farming systems. Interestingly, the highest positivity was observed for tetracycline genes in antibiotic-free flocks, raising some concerns about the role of alternative farming systems in the reduction in antibiotic resistance in food-producing animals.

**Abstract:**

Antimicrobial resistance is a complex and widespread problem threatening human and animal health. In poultry farms, a wide distribution of resistant bacteria and their relative genes is described worldwide, including in Italy. In this paper, a comparison of resistance gene distribution in litter samples, recovered from four conventional and four antibiotic-free broiler flocks, was performed to highlight any influence of farming systems on the spreading and maintenance of resistance determinants. Conventional PCR tests, targeting the resistance genes related to the most used antibiotics in poultry farming, along with some critically important antibiotics for human medicine, were applied. In conventional farms, n. 10 out of n. 30 investigated genes were present in at least one sample, the most abundant fragments being the *tet* genes specific for tetracyclines, followed by those for aminoglycosides and chloramphenicol. All conventional samples resulted negative for colistin, carbapenems, and vancomycin resistance genes. A similar trend was observed for antibiotic-free herds, with n. 13 out of n. 30 amplified genes, while a positivity for the *mcr*-1 gene, specific for colistin, was observed in one antibiotic-free flock. The statistical analysis revealed a significant difference for the *tet*M gene, which was found more frequently in the antibiotic-free category. The analysis carried out in this study allowed us to obtain new data about the distribution of resistance patterns in the poultry industry in relation to farming types. The PCR test is a quick and non-expensive laboratory tool for the environmental monitoring of resistance determinants identifying potential indicators of AMR dissemination.

## 1. Introduction

Food-producing animals have been recognized as one of the major sources of antibiotic-resistant pathogenic and commensal bacteria, which may then be transferred to humans via multiples routes, including food chains [1].

In poultry farming, the repeated use of antibiotics, usually administered in feed or drinking water, can lead to the selection of resistant bacteria, especially at the enteric level, with consequent dissemination in poultry litter and the potential contamination of meat products [2].

The use of poultry litter as soil fertilizer can enhance the spreading of pathogenic and commensal bacteria carrying antibiotic resistance genes (ARGs) in water and vegetables. Thus, ARGs are potentially transferrable to other microorganisms by mobile elements. In addition, enteric bacteria and their ARGs may be transferred to carcasses during the slaughtering processes (i.e., evisceration) and, consequently, reach final consumers through foodstuff [1].

For this reason, in recent years, growing concerns have been expressed by international health agencies and consumers about the transmission of antibiotic-resistant bacteria to humans [3]. As a result, manufacturers in the poultry sector have adopted alternative production systems based on the reduced use of antibiotics, including organic production and antibiotic-free lines [4].

In Italy, the popularity of alternative systems, which were only recently introduced, is fast growing: they already account a 20% increase in product consumption during 2020/2021 [5]. Consumer perceptions that organic/antibiotic-free chicken is healthier and safer than conventional poultry and “does not contain antibiotics” are driving this continuous and impressive increase [6]. Despite the economic importance of these poultry productions, there is no systematic data collection or analysis concerning the presence of resistant bacteria in these systems, and therefore, the effects of these alternative productions on the development of antibiotic resistance in poultry are being studied.

Antibiotic resistance profiles have been investigated in different bacterial species such as *Escherichia coli*, *Salmonella* serovars and *Campylobacter* spp., comparing organic, antibiotic-free, and conventional poultry farming systems with non-univocal results. Indeed, conventional farms generally showed the highest number of resistant bacteria, but raising methods without the use of antibiotics may not be effective to reduce antimicrobial resistance in poultry litter [7,8,9,10].

More recently, a culture-independent approach has been applied to investigate the ARGs distribution in chicken intestinal microbiota, performing PCR-based or metagenomic sequence analysis [11,12,13]. These alternative methods appear to be particularly useful to study the microbial community and its genetic profiles in different types of samples, such as animal manure or agricultural soils, providing more extensive information on the resistance patterns harbored by all bacteria from investigated samples and not only in selected colonies [11,12].

Therefore, few data about the effects of alternative farming systems on ARGs in the environment or poultry products are available.

A pilot study, carried out to investigate the intestinal microbioma and ARGs from conventional and antibiotic-free chickens, highlighted a significantly lower antimicrobial resistance load in antibiotic-free production lines. However, this evidence disappeared when the corresponding carcasses were compared [14]. Conversely, Salerno et al. [15] demonstrated that the abundance of ARGs harbored by the microbial community coming from broilers raised without antibiotics can be considered comparable to what has been reported in conventional farms.

Based on these data, Italian poultry farms of both conventional and antibiotic-free systems are included in the present work, with the aim to investigate the distribution of selected ARGs in litter samples by means of end-point PCR screening, and to highlight the influence of rearing chicken models on the resistance patterns in poultry farms. The resistance determinants related to the most used antibiotics in veterinary practice along with some highly/critically important antibiotics for human medicine are included in order to investigate the occurrence of emergent resistance patterns in the poultry industry.

## 2. Materials and Methods

### 2.1. Sampling Activities

The litter sampling activities involved n. 4 conventional (C1–C4) and n. 4 antibiotic-free (AF1–AF4) intensive broiler flocks, all belonging to the same poultry meat production chain located in central Italy (Abruzzi region). All the investigated farms had a production cycle of about 35–50 days, with a density ranging from 12.000 to 28.000 animals (Ross and Hubbard genetic lines).

To make the samples as homogeneous as possible, n. 5 subsamples of litter were collected at different points in the shed (at the center and four corners of the house) for each flock at two different sampling times: T0 at 7–10 days of animals’ age and T1 at the end of the cycle before the slaughtering, for a total of n. 16 samples. Samples were homogenized to 10% *w*/*v* in 25 mL of sterile physiological solution, mixed appropriately with a Stomacher (VWR International PBI, Milan, Italy), and heated at 75 °C for 20 min to kill bacterial vegetative cells and prevent further multiplication. After that, 300 μL of each solution was used to extract DNA.

### 2.2. Extraction of Nucleic Acids and ARG Screening

Total DNA was recovered from each sample by means of the Maxwell^®^ 11 Instrument (Promega, Italy) using the related Maxwell kit^®^ 11 Tissue DNA Purification (Promega, Italy), allowing us to obtain 100 μL of high-quality DNA useful for the PCR testing. Previously published primers, targeting specific ARGs fragments, were selected in order to perform single or multiplex end-point PCR protocols, as already described [11]. In particular, ARGs commonly used as antibiotics in poultry farming for therapeutic purposes (*tet*A, *tet*B, *tet*C, *tet*K, *tet*L, *tet*M, *tet*B(P), *tet*A(P) for tetracyclines, *lnu*A, *lnu*B for lincomycin, and *aad*A2, *aad*B, *aac*(3)IV for aminoglycosides) [16], along with some highly/critically important antimicrobials for human medicine (*cat*A1 for chloramphenicol, *mcr*-1 to *mcr*-5 for colistin, *van*A, *van*B, *van*C1, *van*C2, *van*D, *van*M, *van*N for vancomycin, and *IMP*, *OXA-48*, *NDM*, *KPC*for carbapenems) [17], were investigated. For each PCR protocol, DNA obtained from resistant bacterial strains and sterile distilled water were added as positive and negative controls.

### 2.3. Statistical Analysis

To highlight any differences in the distribution of ARGs between the two types of farming systems (conventional and antibiotic-free), the exact Fisher test for the analysis of each target gene and the χ2 test for the evaluation of the antimicrobial class resistance were applied using the standard statistical software package STATA [18]. Statistically significant results were considered when *p* < 0.05.

## 3. Results

All tested samples resulted positive for at least one of the investigated ARGs. In conventional farms, the most amplified ARGs were related to the tetracyclines, aminoglycosides, and chloramphenicol resistances, with the *aad*A2 gene recovered from all litter samples, followed by the *cat*A1, found in C1-C3 flocks, while six out of eight *tet* genes (*tet*A, *tet*B, *tet*K, *tet*L, *tet*M, *tet*A(P)) were present in almost two C flocks. Finally, the *lnu* genes, specific against the lincomycin, were recovered from C2 and C3 flocks. The highest number of ARGs was observed in samples collected from C3 flock. All conventional litter samples were negative for colistin, vancomycin, and carbapenem ARGs, and for the *aad*B and *aac*(3) IV genes responsible for aminoglycoside resistance (Figure 1 and Table 1).

All tetracycline resistance genes were detected in AF herds with the following distribution: *tet*A, *tet*B, and *tet*M were amplified in all farms, followed by the remaining tet target fragments, which were recovered from at least two out of four AF flocks. Besides the *tet* genes, the most amplified ARG was the *aad*A2 fragment, specific for aminoglycosides. Only the lincomycin *lnu*A resistance gene was present in AF1 and AF4 flocks; on the contrary, no amplification of the *lnu*B gene was obtained from any antibiotic-free samples. The *cat*A1 resistance gene was amplified in AF1 and AF2 flocks, while the *mcr*-1 gene, specific for colistin resistance, was observed in the AF1 farm. Moreover, the AF1 flock showed the highest frequency of ARGs. Similar to what was observed in the conventional group, all AF samples tested negative for carbapenems and vancomycin resistance genes (Figure 1 and Table 1).

Considering the two times of sampling (T0 and T1), a mild increase in tetracyclines and lincomycin-related genes, for both types of farming, was observed at T1 (Figure 2).

The analysis of the results did not reveal any significant difference, except in the tetracyclines class, where resistance genes were found more frequently in the antibiotic-free category. In more detail, the *tet*M gene was more widely distributed in the antibiotic-free farming line compared to the conventional one (*p* < 0.05).

## 4. Discussion

Antimicrobial resistance is a complex and widespread problem that threatens human and animal health, the global economy, and national and global security [19]. As a result, the poultry industry has adopted alternative production systems based on a reduced use of antibiotics, including organic production and the antibiotic-free line [4]. The effects of these alternative production systems on the development of antibiotic resistance in poultry are, therefore, being studied. For this reason, both conventional and antibiotic-free Italian poultry farms have been included in the present work, helping to improve our knowledge on the environmental diffusion of ARGs in broiler manure. In addition, a culture-independent method has been applied in order to obtain a rapid and more comprehensive analysis of the effective distribution of resistance patterns in broiler farming, analyzing each flock at two different points of the cycle. The use of a PCR-based screening of ARGs, without any microbiological analysis, allowed the detection of several antimicrobial resistance patterns regardless of the bacteria species involved, resulting in particularly useful-to-analyze environmental samples, such as the poultry litter under study.

A similar survey has already been performed in central Italy, involving only conventional broiler and turkey flocks, while a more extensive study was conducted in a high-density farming area of northern Italy, focusing on the microbial community and ARGs abundance in different types of conventional livestock manure (dairy cattle, poultry, and swine) [11,12]. More recently, a single antibiotic-free broiler farm, located in northern Italy, was investigated to quantify some representative ARGs at different sampling sites [15]. Finally, a metagenomic approach, with the aim to analyze the cecal resistome of broiler carcasses at the slaughterhouse, was performed by De Cesare et al. [14]. In this respect, our study represents a first attempt to compare the distribution of different ARGs against several classes of antibiotics between conventional and antibiotic-free poultry farming systems in Italy, sampling the fecal specimens at the flock level and applying a rapid, low-cost, and relatively easy-to-perform method.

Based on the results obtained by the PCR screening, a wide diffusion of ARGs in both types of sampled farms has been demonstrated, with slight differences in the class of investigated antibiotics and the type of production cycle.

The most detected genes in both groups were related to tetracycline resistance, probably due to their wide use in veterinary practice. In 2020, official data showed that Italy still represented the EU country with the highest proportion of antibiotic sales for food-producing animals, with particular emphasis on penicillins and tetracycline molecules [16]. However, the most evident reduction in tetracyclines sales was achieved in Italy, compared to other EU countries, considering the 2010–2020 trend [16], and in recent years, the poultry production chain involved in the sampling activities has considerably reduced the use of tetracyclines for therapeutic treatments, excluding these types of molecules in conventional broiler flocks [20].

Probably, the wide diffusion of *tet* genes observed in this study is consequent to previous recurrent treatments that have established long-term selective resistant bacteria harboring these genetic elements in the farm environments [21]. Moreover, other environmental sources of resistance pollution by genetic determinants (water, food, or wild birds) cannot be ruled out [22,23]. Indeed, the *tet* positivity suggests a wide and multi-factorial dissemination of these resistance determinants in poultry flocks, in accordance with what has been reported in other countries such as Portugal [24], Tunisia [25], the USA [26], and China [27]. In this respect, the increase in tetracycline genes amplified in T1 samples observed in both types of farming systems should be related to multiple sources of contamination of resistance determinants. Probably, the chickens were exposed to other extra-intestinal resistant bacteria during the cycle production, as already suggested [15].

The comparison between conventional and antibiotic-free flocks revealed a significant difference for *tet* gene distribution, being more amplified in antibiotic-free samples, with particular regard to the *tet*M gene. This does not imply, however, that there is a greater risk of transmission of these genes from animals to humans; this aspect, in fact, remains debated and controversial. The hypothesis that the small number of investigated samples could have influenced this result cannot be ruled out, as well as the use of conventional end-point PCR, which could have influenced the results. Indeed, the detection limit of applied protocols may not have been sufficiently high to amplify the target fragments in all samples, while alternative methods, such as quantitative PCR, should be preferred for further investigations.

A recent study about the spread of ARGs in fecal poultry samples, collected from layer and broiler antibiotic-free farms, has shown a high positivity for the *tet*M gene, probably due to the genetic transfer by bacterial species naturally present in the fecal microbiota of the chickens, or because it was already widely spread in the environment and thus unrelated to previous antibiotic treatments in animals or humans [28].

Similar results were observed in the USA for the *tet*A gene in pastured “no antibiotic ever” poultry farms, showing a greater abundance than in cattle or sheep manure [29], and in Italy, where the environmental fecal samples recovered from a broiler antibiotic-free farm revealed a wide distribution of *tet*A [15].

Finally, *tet* genes are related to numerous bacterial species of chicken gut microbiota, as recently observed in commensal isolates recovered from healthy animals and analyzed by whole-genome sequencing [30]. Several *tet* genes are often associated with moving elements in bacteria, such as the transposon Tn 916-like in *Bacillus cereus* of swine origin [31], and in *Staphylococcus aureus* from different animal species including poultry [32]. However, comparing bacterial isolates coming from the two types of farming, the results available in the literature appear to be opposite or not conclusive, such as enterococci, which showed similar antibiotic resistance profiles in both conventional and antibiotic-free broilers [33], while resistant *E. coli* was less frequently isolated from alternative farming models, including antibiotic-free poultry production [34].

Concerning the other resistance genes, the *aad*A2 gene, specific for aminoglycosides, was the most representative gene in samples from all the investigated farms, confirming a previous study carried out on the same geographical area [11] and in accordance with the aminoglycoside resistance profiles described in bacterial isolates coming from poultry farms in Italy [35].

The *lnu*A and *lnu*B genes, both coding for lincosamide nucleotidiltransferase enzymes (responsible for lincomycin resistance), were slightly distributed in the flocks under study, with a higher positivity for the *lnu*A gene compared to the *lnu*B gene. Indeed, lincomycin-resistant strains have been described in several genera of poultry origin, such as *Clostridium*, *Campylobacter*, and *Salmonella* [36,37,38].

The *cat*A1 gene appeared to be more detectable in conventional flocks. The *cat*A1 gene is often associated with plasmids, transposons, and gene cassettes [39], and has recently been amplified in *Salmonella enterica* pRH-R11 and *E. coli* pRH-R111 plasmids coming from livestock in Germany [40]. Although the use of chloramphenicol in zootechnical species has been banned in Europe since 1994 due to its toxicity and potentially adverse effects resulting from residues in food-producing animals [41], resistant bacteria continue to harbor the relative gene [42]. The presence of the *cat*A1 gene highlighted in this investigation could be related to the same plasmid also carrying the *aad*A2 gene, the latter chosen as a target for resistance to aminoglycosides, as previously demonstrated by the evidence of the association between ARGs against chloramphenicol and aminoglycosides [43].

Interestingly, the *mcr*-1 gene, encoding the enzyme phosphoethanol amine transferase, responsible for polymyxin and colistin resistance, was found only in two samples from antibiotic-free herds, while conventional herds were found to be negative. Despite the fact that antibiotic resistance to colistin is considered emergent, as indicated in studies performed in Tunisia [44], but especially in China and Vietnam [27,45], the low prevalence observed in the analyzed poultry farms suggests a negligible risk for poultry meat consumers. In accordance with this, a reduction trend of *mcr*-1 circulation in human *Enterobacterales* isolates was reported in Italy [46]. Recently, the *mcr*-1 has been found at the plasmid level in *Salmonella* Enterica and *E. coli* in poultry farms in Portugal [47], and a systematic review revealed a wide distribution of *mcr* genes in all continents, being greater in Asia, followed by Europe, Africa, and the Americas [48]. In Italy, the *mcr*-1 gene was significantly more abundant in swine farms compared with chicken flocks [12]. However, additional investigations could be carried out in order to monitor any epidemiological changes in emergent *mcr* genes diffusion in poultry farming and to identify potential sources of colistin resistance pollution in non-human environments.

No samples tested positive for vancomycin resistance genes, in line with the limitation of its use since 1997, following a Danish study showing that the use of avoparcin (belonging to the same class as vancomycin) as a growth promoter was associated with the selection of vancomycin-resistant strains of *Enterococcus faecium* in poultry [49]. Despite this, poultry farming is still considered a potential reservoir of vancomycin resistance, but the results obtained by this work suggest a decline in this trend. Indeed, in a recent German study, only one strain of *Enterococcus faecium*, carrying the *van*A gene, was isolated from poultry carcasses [50].

Finally, no samples resulted positive for carbapenem resistance genes, in line with other European countries [51].

## 5. Conclusions

Even if not representative of the resistome, the analysis carried out in this study was focused on the ARGs related to the most common antibiotics used in the poultry industry along with some antimicrobials considered critical for human health, investigating the potential influence of alternative farming systems on their dissemination. The results did not allow us to obtain conclusive evidence that antibiotic-free farming can be an effective model to reduce the maintenance of ARGs, suggesting that other determinants, related to the environment and/or previous antibiotic uses in animals and humans, should be considered. The culture-independent PCR protocols applied in our investigations allowed us to quickly obtain data, which was useful to evaluate the environmental distribution of recurrent or emergent AMR indicators, and could be an additional tool for monitoring activities.

## Figures and Tables

**Figure 1 animals-12-02310-f001:**
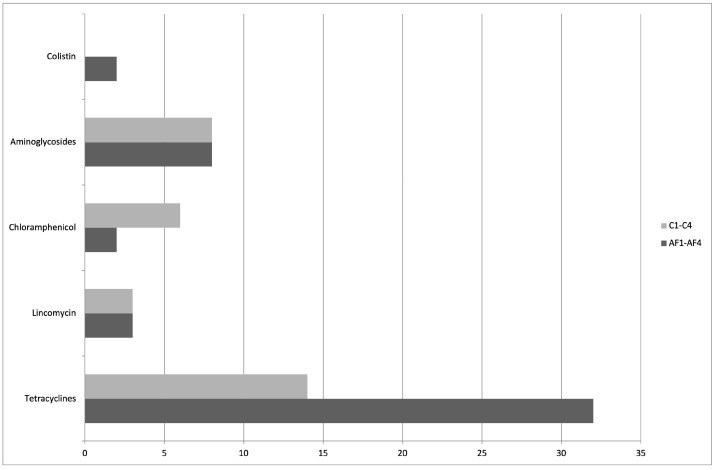
Total ARGs fragments amplified by PCR test, grouped by antibiotic classes and types of farming. The frequency of all target fragments was obtained by summing the number of PCR tests that returned a positive result from each ARG and litter sample.

**Figure 2 animals-12-02310-f002:**
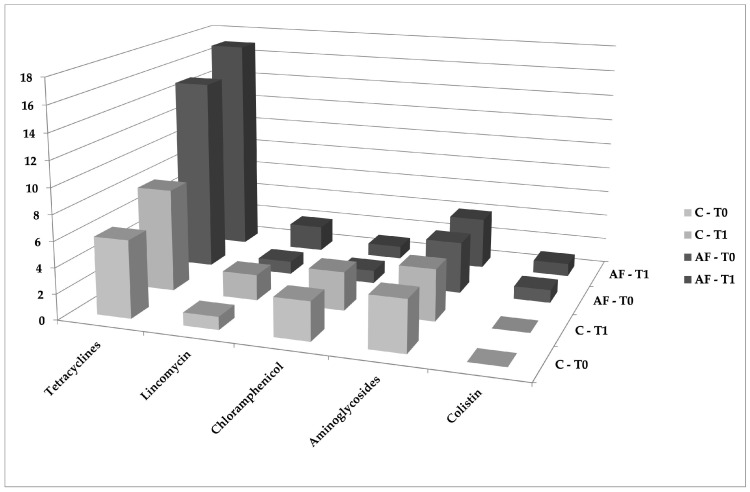
Temporal distribution of ARGs detected in T0 and T1 samples, grouped by antibiotic classes and types of farming. The frequency of all target fragments was obtained by summing the number of PCR tests that returned a positive result from each of the ARGs and litter samples.

**Table 1 animals-12-02310-t001:** Distribution of investigated ARGs in conventional and antibiotic-free flocks.

Farms	ARG Litter Samples
C1	*tet*A, *tet*B, *tet*L, *cat*A1, *aad*A2
C2	*tet*A, *tet*B, *tet*L, *cat*A1, *aad*A2*, lnu*A
C3	*tet*K, *tet*M, *tet*A(P), *cat*A1, *aad*A2*, lnu*B
C4	*tet*K, *tet*M, *tet*A(P), *aad*A2
AF1	*tet*A, *tet*B, *tet*C, *tet*K, *tet*L, *tet*M, *lnu*A, *cat*A1, *aad*A2, *aac*(3)IV, *mcr*-1
AF2	*tet*A, *tet*B, *tet*K, *tet*L, *tet*M, *tet*A(P), *tet*B(P), *cat*A1, *aad*A2
AF3	*tet*A, *tet*B, *tet*M, *tet*A(P), *tet*B(P), *aad*A2, *aac*(3)IV
AF4	*tet*A, *tet*B, *tet*C, *tet*M, *Inu*A, *aad*A2

## Data Availability

Data supporting results presented in this study can be acquired from the corresponding author.

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
