# Peer review of "Antibiotic Resistance Genes Occurrence in Conventional and Antibiotic-Free Poultry Farming, Italy"

_animals, 2022, doi:10.3390/ani12182310_

Round 1

Reviewer 1 Report

The Introduction would benefit strongly from adding a section dealing with the risks of the presence of antibiotic resistant bacteria in the poultry-related products and materials (e.g. litter, which was examined in this study). Please elaborate more on how and where these bacteria and antibiotic resistance genetic determinants spread and what are the risks associated with this spread.

Why do the Authors use only PCR-based approach? Comparing the PCR-based with culture-based results would definitely tell us more about the possible risks associated with antibiotic resistance.

Materials and Methods

Sampling: If I understand it correctly, the Authors sampled a total of 8 broiler flocks, twice? This makes a total of 16 samples? If this is correct, the number of analyzed samples is much too small to draw any substantiated conclusions. Either elaborate more on the sampling procedure, or conduct additional experiments to make your results publishable. This is even more non-understandable, when you look at Figure 1 (how do you get over 30 positive samples out of 8 flocks?).

l. 109-114: What was the background (merit) for examination of these particular antibiotics? The Authors state that these were crucial antibiotics used in human and veterinary medicine, but what about beta-lactams? These are actually among the most important and widely used antibiotics in human medicine. Please refer to the literature that lead you to the selection of the AR genes.

Please specify whether any positive or negative control was applied in your study? The obtained results (specifically the presence of colistin resistance determinants in antibiotic-free flocks) are puzzling. Colistin is a strictly hospital administered, last resort antibiotic, and the common spread of the resistance mechanism to this antibiotic seems not only worrying but worth detailed investigation.

Results

Figure 1 is uncomprehensive. What does it mean e.g. that you had over 30 positive samples for tetracyclines for AF group? This needs to be clearly stated, I suggest in the Figure caption.

l. 148-149: how would you explain an increase in tet genes in antibiotic-free farming system?

l. 157-158: no surprise that the statistical analysis did not reveal any significant differences when you test a total of 16 samples!

Discussion

l. 180 both cited papers (11 and 12) present research conducted on a larger number of samples (over 30).

The fact that the antibiotic-free system was characterized by higher abundance of AR genes has actually not been discussed. What can be the reason for such observation? My suggestion is – too small number of samples examined.

You cannot state that the presence of colistin resistance determinant is not worrying, because you detected it in two samples. This is for sure worrying. Or should be more thoroughly investigated.

Conclusions

l. 276-279: actually, your study showed the opposite.

Author Response

The Introduction would benefit strongly from adding a section dealing with the risks of the presence of antibiotic resistant bacteria in the poultry-related products and materials (e.g. litter, which was examined in this study). Please elaborate more on how and where these bacteria and antibiotic resistance genetic determinants spread and what are the risks associated with this spread.

In accordance with the Reviewer’s suggestion, we added a sentence about the AMR transmission routes from the poultry litter to the final consumer (lines 54-59).

Why do the Authors use only PCR-based approach? Comparing the PCR-based with culture-based results would definitely tell us more about the possible risks associated with antibiotic resistance.

We are grateful to the Reviewer for this comment but the paper aimed to investigate the ARGs diffusion in litter specimens regardless of bacterial species isolated by means of culture methods. Indeed, traditional microbiology allows to investigate only a minor part of entire enteric microbiota focusing on the resistance profiles of selected colonies. On the contrary, the PCR-based approach has been applied in our study with the aim to determine the diffusion of resistance genetic elements that can be harbored by different and numerous microorganisms (cultivable and uncultivable bacteria), without limiting our investigations to a few bacterial species. For examples, it has been estimated that in human the 30% of the entire gut bacteria are currently cultured and investigated (Liu et al., Enlightening the taxonomy darkness of human gut microbiomes with a cultured biobank. Microbiome. 2021 doi: 10.1186/s40168-021-01064-3). In order to highlight the advantages of biomolecular methods, we added a sentence in the introduction section (lines 82-86).

Materials and Methods

Sampling: If I understand it correctly, the Authors sampled a total of 8 broiler flocks, twice? This makes a total of 16 samples? If this is correct, the number of analyzed samples is much too small to draw any substantiated conclusions. Either elaborate more on the sampling procedure, or conduct additional experiments to make your results publishable.

We appreciated the comment provided by the Reviewer, anyway we should point out that the study carried out in our paper focused on the environmental diffusion of ARGs in litter coming from poultry flocks. Similarly to what performed in other cited studies (Di Francesco et al., 2021; Laconi et al., 2020; Salerno et al., 2022), the number of environmental samples collected from each flock is not particularly high, because the samples are not representative of the number of raised animals but they should represent the environment where the animals live and spread their intestinal microbiota. In particular, Laconi et al. (Sci Total Environ. 2021, 760:143404) collected for each type of farm only one sample of manure and n. 10 conventional poultry farms have been investigated, for a total of 10 manure samples. Salerno et al. (Poult Sci. 2022, 101(3):101675) reported the PCR analysis carried out from one antibiotic-free farm collecting three fecal specimens at different raising times (2 wk old, 60 day old and after depopulation). In line with these data, in our study we collected two different litter samples (at 7-10 day old and before slaughtering) from each flock, for a total of 16 samples. Each sample was composed by 5 pools of fecal material collected from 5 different points of the chickens house. In order to better clarify the sampling methodology we rewritten the description of sample composition in the Sampling activities paragraph, in agreement with the Reviewer 3, too (lines 112-115).

This is even more non-understandable, when you look at Figure 1 (how do you get over 30 positive samples out of 8 flocks?).

We realized that the legend of the Figure 1 is not completely appropriate to describe the results. The number reported in the figure is related to the total of ARGs target fragments amplified starting from the samples under study. In fact, for each antibiotics class different ARGs have been investigated by PCR (as detailed in the Material and methods section, 8 tet genes for tetracyclines, 2 Inu genes for lincomycin…) for a total of 30 different genes target. Therefore, the Figure 1 shows the frequency of amplified ARGs fragments (each fragment has been amplified in one or both litter samples collected from each flock), grouped by antibiotic classes. Accordingly, the Figure 1 legend has been properly corrected and the image has been replaced with a higher quality one (600 dpi), as suggested by the Reviewer 2.

  1. 109-114: What was the background (merit) for examination of these particular antibiotics? The Authors state that these were crucial antibiotics used in human and veterinary medicine, but what about beta-lactams? These are actually among the most important and widely used antibiotics in human medicine. Please refer to the literature that lead you to the selection of the AR genes.

We are conscious that the molecules included in this paper are not representative of all classes considered important for veterinary medicine and human health. However, considering the available economic resources, the selection of targeted ARGs was performed based on the following criteria:

  • The official data available about the sales of antibiotics for veterinary use registered in Italy by the European Surveillance of Veterinary Antimicrobial Consumption (2021),
  • The results obtained by a previous survey carried out in the same area of Italy (Di Francesco et al., 2021).
  • The availability of standardized PCR protocols routinely applied in our laboratory.

Regarding the antibiotics for human medicine we decided to focus on the critically important antibiotics exclusively used for human therapy (colistin, chloramphenicol, vancomycin and carbapenems) based on the classification of WHO (2019).

In agreement with the Reviewer, we added the references considered for the ARGs selection.

Please specify whether any positive or negative control was applied in your study?

We added the positive and negative controls used for PCR amplification (lines 132-133).

The obtained results (specifically the presence of colistin resistance determinants in antibiotic-free flocks) are puzzling. Colistin is a strictly hospital administered, last resort antibiotic, and the common spread of the resistance mechanism to this antibiotic seems not only worrying but worth detailed investigation.

We agree with the Reviewer about the worrying recovery of mcr-1 gene in animal specimens, however the low number of positive samples obtained in our study, compared with the most dramatic diffusion, not only of mcr-1 but also of other additional colistin resistance determinants (from mcr-2 to mcr-10) in both human and animals sources, with particular regard for China, (Wang et al., More diversified antibiotic resistance genes in chickens and workers of the live poultry markets. Environ Int. 2021 Aug;153:106534. doi: 10.1016/j.envint.2021.106534), allowed to suggest that the trend of mcr genes diffusion in poultry farming system is decreasing.

Results

Figure 1 is uncomprehensive. What does it mean e.g. that you had over 30 positive samples for tetracyclines for AF group? This needs to be clearly stated, I suggest in the Figure caption.

Done

  1. 148-149: how would you explain an increase in tet genes in antibiotic-free farming system?

As suggested in Discussion section, other environmental sources of resistance pollution by genetic determinants (water, food or wild birds) may be considered responsible of the wide dissemination of tet genes, harbored by numerous bacterial species, including commensal and environmental microorganisms (lines 199-204). Probably, these potential sources may have influenced the temporal increase of resistance genes, observed in T1 samples and regardless of the farming system (antibiotic-free and conventional). This hypothesis has been added in the discussion section (lines 226-229).

  1. 157-158: no surprise that the statistical analysis did not reveal any significant differences when you test a total of 16 samples!

Indeed, a possible explanation of this result can be related to the sample size. Therefore, the statistical method applied in our study (exact Fisher test) is considered suitable when the sample size is small (Kim HY. Statistical notes for clinical researchers: Sample size calculation 2. Comparison of two independent proportions. Restor Dent Endod. 2016 May;41(2):154-6. doi: 10.5395/rde.2016.41.2.154).

Discussion

  1. 180 both cited papers (11 and 12) present research conducted on a larger number of samples (over 30).

As reported above, both papers highlighted by the Reviewer investigated the ARGs diffusion analyzing a number of litter specimens similar to what collected in the manuscript. Despite the small number of collected samples the statistical analysis was performed comparing the frequency of detected ARGs among samples or animal species (Laconi et al., 2020).

The fact that the antibiotic-free system was characterized by higher abundance of AR genes has actually not been discussed. What can be the reason for such observation? My suggestion is – too small number of samples examined.

We added a sentence in discussion section to include this hypothesis, along with a comment about the detection limit of end-point PCR applied, as suggested by the Reviewer 2 (lines 234-239).

You cannot state that the presence of colistin resistance determinant is not worrying, because you detected it in two samples. This is for sure worrying. Or should be more thoroughly investigated.

We agree with the Reviewer about the worrying recovery of mcr-1 gene in animal specimens, however the low number of positive samples obtained in our study, compared with the most dramatic diffusion, not only of mcr-1 but also of other additional colistin resistance determinants (from mcr-2 to mcr-10) in both human and animals sources, with particular regard for China, (Wang et al., More diversified antibiotic resistance genes in chickens and workers of the live poultry markets. Environ Int. 2021 Aug;153:106534. doi: 10.1016/j.envint.2021.106534), allowed to suggest that the trend of mcr genes diffusion in poultry farming system is decreasing.

However, we added a sentence to suggest further investigations to determine the origin and the level of dissemination of colistin resistance patterns in food producing animals (lines 292-294).

Conclusions

  1. 276-279: actually, your study showed the opposite.

We believe that this sentence cannot be excluded considering the similar dissemination of ARGs observed in both farming systems, except for the tetM gene, resulted more frequent in AF model, as supported by the statistical analysis. In addition, despite the small size of investigated samples, the results are in lines with similar studies.

Reviewer 2 Report

Antimicrobial resistance (AMR) is a complex and widespread problem that threatens human and animal health. It is true that the use of antimicrobials in livestock exerts a selective pressure on the resistance determinants (genes) and their persistence in the intestinal microbiota. Furthermore, poultry farming really represents a potential driver of such genes, as they can reach the environment or the food chain from the litter. The manuscript describes a comparison of the distribution of resistance genes in litter samples, recovered from 4 flocks of conventional broilers and 4 without antibiotics, trying to highlight the influence of farming systems on the spread and maintenance of resistance determinants.

This kind of study is necessary to obtain new data on the distribution of resistance patterns in poultry farms. The research question is very important to demonstrate the occurrence of antimicrobial resistance in any poultry production chain (broilers, layers, breeders). However the manuscript is still not well-prepared, having many flaws that prevent its publication. Furthermore, it is more dramatic than technical, as the few results obtained are not enough to show a wide diffusion of resistance genes in chicken litter from both farming systems. I'm also not very enthusiastic about detcetion of genetic determinants in environmental samples without bacterial isolation. I will point out this and other concerns in the next paragraphs.

First, the title is not true, as the manuscript does not demonstrate the wide spread of AMR in Italy. Only a comparison with a few samples is reported. Therefore, I recommend that authors be more honest in the title. Please also review the statement that “The results showed a wide spread of resistance genes in poultry litter from both farming systems” in the Simple Summary.

The introduction looks ok. I agree with the authors that it is very important to inform consumers that organic/antibiotic-free chicken is not always healthier and safer than conventional chicken. And it is very important to report that it may “contain antibiotics”. However, the entire Introduction could be improved by including more scientific data, mainly describing better other scientific studies that investigate the genetic determinants of AMR without isolating the bacteria. Please also replace “level of spreading” by “occurrence” (same concern as commented to the title).

In Materials and Methods, sampling is poorly described. It is really necessary to explain the farms in more detail, if possible including a map with their location. The collection of litter specimens should also be more detailed. It is also not acceptable to describe “several specimens of litter”. Please inform more details! The authors have already described this procedure more clearly  (11). However, my main concern is the use of conventional PCR, especially procedures that amplify large fragments (> 500 bp) that are difficult to evaluate. Why were real-time PCR assays not used? These assays would provide quantitative information. Authors need to include at least one paragraph about the limitation of their PCR procedures in the Discussion.

I'm also very concerned about the Results. Tables and Figures could be better prepared, with more informative illustrations, as well as higher quality images. I think the number of tables could be reduced after this improvement. I also suggest that the authors be more modest in comparing the two farming systems. Finally, the Discussion needs a complete revision and a better structure. Please also include a paragraph discussing the various limitations of the study.

Author Response

Antimicrobial resistance (AMR) is a complex and widespread problem that threatens human and animal health. It is true that the use of antimicrobials in livestock exerts a selective pressure on the resistance determinants (genes) and their persistence in the intestinal microbiota. Furthermore, poultry farming really represents a potential driver of such genes, as they can reach the environment or the food chain from the litter.

The manuscript describes a comparison of the distribution of resistance genes in litter samples, recovered from 4 flocks of conventional broilers and 4 without antibiotics, trying to highlight the influence of farming systems on the spread and maintenance of resistance determinants.

This kind of study is necessary to obtain new data on the distribution of resistance patterns in poultry farms. The research question is very important to demonstrate the occurrence of antimicrobial resistance in any poultry production chain (broilers, layers, breeders). However the manuscript is still not well-prepared, having many flaws that prevent its publication. Furthermore, it is more dramatic than technical, as the few results obtained are not enough to show a wide diffusion of resistance genes in chicken litter from both farming systems. I'm also not very enthusiastic about detection of genetic determinants in environmental samples without bacterial isolation. I will point out this and other concerns in the next paragraphs.

First, the title is not true, as the manuscript does not demonstrate the wide spread of AMR in Italy. Only a comparison with a few samples is reported. Therefore, I recommend that authors be more honest in the title. Please also review the statement that “The results showed a wide spread of resistance genes in poultry litter from both farming systems” in the Simple Summary.

We appreciated the valuable comments of the Reviewer, however the title didn’t intend to highlight “a wide spread of AMR” but only the evidence that the ARGs can be spread by both antibiotic-free and conventional poultry farms. In order to specify the area of study we added the term “Italy” at the end of the title. As suggested by the Reviewer afterwards we replaced the term “spreading” with “occurrence” and, to respect the meaning of the title, in the Simple Summary we rewritten the sentence as following: “The results showed the presence of several resistance genes in poultry litter of both farming systems” (lines 21-22).

The introduction looks ok. I agree with the authors that it is very important to inform consumers that organic/antibiotic-free chicken is not always healthier and safer than conventional chicken. And it is very important to report that it may “contain antibiotics”. However, the entire Introduction could be improved by including more scientific data, mainly describing better other scientific studies that investigate the genetic determinants of AMR without isolating the bacteria.

As suggested by the Reviewer, the studies regarding the ARGs distribution in conventional and antibiotic-free have been better described (lines 89-95).

Please also replace “level of spreading” by “occurrence” (same concern as commented to the title).

Done

In Materials and Methods, sampling is poorly described. It is really necessary to explain the farms in more detail, if possible including a map with their location.

We are really sorry, but we are not able to produce a geographical map of the flocks under study within the deadline required by the Editor for the submission of revised manuscript (17th August). In alternative, we added the name of Italian region where the farms are located (line 109). Regarding other information, can the Reviewer specify what kind of data are necessary?

The collection of litter specimens should also be more detailed. It is also not acceptable to describe “several specimens of litter”. Please inform more details!

The authors have already described this procedure more clearly  (11).

The sampling was better described as suggested by the Reviewer 1 and 3, too (lines 112-115). The reference (11) was not reported in sampling description because the procedure used in the current study is different.

However, my main concern is the use of conventional PCR, especially procedures that amplify large fragments (> 500 bp) that are difficult to evaluate. Why were real-time PCR assays not used? These assays would provide quantitative information.

We agree with the Reviewer about the advantages derived by the use of quantitative PCR and the lower sensitivity of conventional PCR. Indeed, the evaluation of relative abundance of representative ARGs in poultry flocks is currently in progress at our laboratories. This manuscript represents a preliminary investigation, low-cost and useful to identify the more suitable ARGs fragments that will be used for the next analysis.

Authors need to include at least one paragraph about the limitation of their PCR procedures in the Discussion.

In accordance with the Reviewer, we added a sentence about the limit of our protocols (lines 234-239).

I'm also very concerned about the Results. Tables and Figures could be better prepared, with more informative illustrations, as well as higher quality images.

As suggested by the Reviewer n. 1 and n. 2 we replaced both images with new figures with a higher quality (600 dpi). The legend of Figure 1 was modified to better clarify the distribution of ARGs fragments in samples under study.

I think the number of tables could be reduced after this improvement.

To respect this suggestion, we decided to delete the Table n. 2,

I also suggest that the authors be more modest in comparing the two farming systems.

Sorry, but in order to respect this suggestion, the Authors need to receive a more detailed description of the sentences where the comparison appears to be not appropriate. Indeed, the opinion of the Authors is that all comments reported in discussion are related to the results obtained during the investigations and supported by the statistical analysis performed, where it was applied.

Finally, the Discussion needs a complete revision and a better structure.

Similarly to what reported above, we need more details to change the Discussion section in compliance with the Reviewer.

Please also include a paragraph discussing the various limitations of the study.

Done (lines 234-239).

Reviewer 3 Report

Antimicrobial resistance is the greatest challenge facing modern medicine. Most of the studies on AMR are carried out by isolating cultivable sentinel bacteria that allow studying the phenotypic pattern of resistance by disk diffusion or microdilution in broth. The use of molecular techniques for the ARGs detection without culturing sentinel bacteria offers a significant improvement in AMR surveillance.

The uploaded article focuses on the detection of resistance genes in samples obtained, on the one hand, from conventional broiler flocks, and on the other from antibiotic-free broiler flocks.

The introduction is well done, with relevant and current references.

The methodology is well structured and detailed.

In my opinion, it may be necessary to indicate the number of samples collected at each time and farm, since the figures show the number of positive samples. It would be interesting to know the total number of samples and if there were any negatives for all the genes tested.

In the statistical analysis, it is necessary to indicate from which p-value a result is considered statistically significant since it could be 0.1, 0.05, or 0.01. It is later known, in the results, that it is 0.05 (line 160), but it must be reflected in the methodology.

It would also be interesting to indicate the statistical software, commercial company, and version, used for the analyses.

The results are very well presented: concise and clear. Just one recommendation: on lines 130 and 140 put both "(Table 1 and Figure 1)" since both express the results of both types of farms.

For the rest, I think it is a very interesting study for the scientific community. Congratulations.

Author Response

Antimicrobial resistance is the greatest challenge facing modern medicine. Most of the studies on AMR are carried out by isolating cultivable sentinel bacteria that allow studying the phenotypic pattern of resistance by disk diffusion or microdilution in broth. The use of molecular techniques for the ARGs detection without culturing sentinel bacteria offers a significant improvement in AMR surveillance.

The uploaded article focuses on the detection of resistance genes in samples obtained, on the one hand, from conventional broiler flocks, and on the other from antibiotic-free broiler flocks.

The introduction is well done, with relevant and current references.

The methodology is well structured and detailed.

In my opinion, it may be necessary to indicate the number of samples collected at each time and farm, since the figures show the number of positive samples. It would be interesting to know the total number of samples and if there were any negatives for all the genes tested.

We are very grateful to the Reviewer for this suggestion. We improved the sampling description in Material and methods section, to describe the composition and the total number of collected samples. In addition, in accordance also with the Reviewers 1 and 2, the legend of Figure 1 has been corrected because the image doesn’t show the number of positive litter samples but the total number of amplified ARGs starting from the n. 16 samples (lines 112-115). Finally, we added a sentence in Results section to specify that “All tested samples resulted positive for at least one investigated ARGs” (line 142).

In the statistical analysis, it is necessary to indicate from which p-value a result is considered statistically significant since it could be 0.1, 0.05, or 0.01. It is later known, in the results, that it is 0.05 (line 160), but it must be reflected in the methodology.

It would also be interesting to indicate the statistical software, commercial company, and version, used for the analyses.

Done (lines 138-139).

The results are very well presented: concise and clear. Just one recommendation: on lines 130 and 140 put both "(Table 1 and Figure 1)" since both express the results of both types of farms.

Done

For the rest, I think it is a very interesting study for the scientific community. Congratulations.

Thank you very much for the positive feedback received by the Reviewer and for the valuable suggestions useful to improve the quality of our manuscript.

Round 2

Reviewer 1 Report

The requested changes have been included.

Author Response

Thank you for your response. We just added some sentences in the Discussion section in agreement with the suggestions of the Reviewer n. 2.

Reviewer 2 Report

I have previously reviewed this manuscript. As I pointed out before, this kind of study is necessary to publish new data on the distribution of resistance patterns in poultry farms. The research question is very important to demonstrate the occurrence of antimicrobial resistance in any poultry production chain (broilers, layers, breeders).

In this new version, the authors have improved the whole manuscript. They have modified the Title and the Simple Summary as suggested. The Introduction and Material and Methods are OK now. The Results section was not improved and I think the authors could prepare even better the Figure 1 and 2 as well as provide more complete and informative subtitles.

In the Discussion section, the authors were less emphatic and more realistic with the few number of samples and limited findings of their study as recommended. However, I really think the authors could discuss even better their results in comparison with other studies. As I highlighted before, I'm not very enthusiastic about detection of genetic determinants in environmental samples without bacterial isolation. I would expect one paragraph discussing this strategy. In addition the authors could compare the detection of the genes by specific PCRs in comparison with other strategies, as for example metagenomics and mass spectrometry (references 12 and 14). So I suggest the authors to reinforce the Discussion to improve even more the manuscript.

Finally an English grammar revision is also necessary. 

Author Response

I have previously reviewed this manuscript. As I pointed out before, this kind of study is necessary to publish new data on the distribution of resistance patterns in poultry farms. The research question is very important to demonstrate the occurrence of antimicrobial resistance in any poultry production chain (broilers, layers, breeders).

In this new version, the authors have improved the whole manuscript. They have modified the Title and the Simple Summary as suggested. The Introduction and Material and Methods are OK now.

The Results section was not improved and I think the authors could prepare even better the Figure 1 and 2 as well as provide more complete and informative subtitles.

As suggested by the Reviewer we added a sentence in the Figure 1 and Figure 2 legends to better specify the data showed in the graphs.

In the Discussion section, the authors were less emphatic and more realistic with the few number of samples and limited findings of their study as recommended. However, I really think the authors could discuss even better their results in comparison with other studies. As I highlighted before, I'm not very enthusiastic about detection of genetic determinants in environmental samples without bacterial isolation. I would expect one paragraph discussing this strategy. In addition the authors could compare the detection of the genes by specific PCRs in comparison with other strategies, as for example metagenomics and mass spectrometry (references 12 and 14). So I suggest the authors to reinforce the Discussion to improve even more the manuscript.

We improved the Discussion section to better elucidate the strategy used in the study (lines 199-208; lines 210-212; lines 214-215).

Finally an English grammar revision is also necessary. 

We checked the English grammar with the assistance of a native speaker.